# 24-Gaze-Point Calibration Method for Improving the Precision of AC-EOG Gaze Estimation

**DOI:** 10.3390/s19173650

**Published:** 2019-08-22

**Authors:** Muhammad Syaiful Amri bin Suhaimi, Kojiro Matsushita, Minoru Sasaki, Waweru Njeri

**Affiliations:** Department of Mechanical Engineering, Gifu University, 1-1 Yanagido, Gifu 501-1193, Japan

**Keywords:** EOG, gaze estimation, affine transformation, cross shaped-electrode, plus-shaped electrode

## Abstract

This paper sought to improve the precision of the Alternating Current Electro-Occulo-Graphy (AC-EOG) gaze estimation method. The method consisted of two core techniques: To estimate eyeball movement from EOG signals and to convert signals from the eyeball movement to the gaze position. In conventional research, the estimations are computed with two EOG signals corresponding to vertical and horizontal movements. The conversion is based on the affine transformation and those parameters are computed with 24-point gazing data at the calibration. However, the transformation is not applied to all the 24-point gazing data, but to four spatially separated data (Quadrant method), and each result has different characteristics. Thus, we proposed the conversion method for 24-point gazing data at the same time: To assume an imaginary center (i.e., 25th point) on gaze coordinates with 24-point gazing data and apply an affine transformation to 24-point gazing data. Then, we conducted a comparative investigation between the conventional method and the proposed method. From the results, the average eye angle error for the cross-shaped electrode attachment is x=2.27°±0.46° and y=1.83°±0.34°. In contrast, for the plus-shaped electrode attachment, the average eye angle error is is x=0.94°±0.19° and y=1.48°±0.27°. We concluded that the proposed method offers a simpler and more precise EOG gaze estimation than the conventional method.

## 1. Introduction

The direct human gaze is an indicator of interest. Thus, eye tracking has become of the most prospective technologies in recent years. There are two types of major eye-tracking systems: One is to record pupil positions with infrared cameras [1,2], and the other is to record electro-occulo-graphic (EOG) signals with a biological signal measurement device [3]. The camera-based system requires the user to wear camera built-in glasses [4,5] or to fix their head position in order to achieve stable capturing of their pupils. This imposes physical burdens, such as heaviness and motion and visual restriction on the user [6,7]. In addition, this method is slightly difficult for users who wear glasses and contact lenses due to reflection from the lenses, which makes the system unable to detect the pupils in a stable manner. Whereas the EOG-based system is characterized by gaze point precision, which is less than that of the camera-based system, the EOG is more robust in determining gaze direction [8,9,10,11,12,13,14]. In the EOG, data acquisition is achieved by attaching a few disposable electrodes around their eye of the subject and has the advantage of being less burdensome to the eyes in the measurement of the eye movement. Considering the aforementioned low accuracy associated with the EOG, it is important to develop techniques in order to improve the accuracy of the EOG-based eye tracker system.

EOG signals are electrical potentials generated when the eyeball moves. This results from the fact that the eyeball acts like a battery, where the eye cornea provides the positive potential and the eye retina provides the negative potential [15]. For that reason, it is possible to estimate eye movements by analyzing the resulting EOG signals. There are two types of EOG analysis methods: Direct-current-EOG (DC-EOG) and alternating-current-EOG (AC-EOG). DC-EOG signals are almost raw EOG signals, and the amplitudes are directly related to the eye movement [16]. DC-EOG has a disadvantage in that the resulting signals are easily influenced by human movement, and motion restriction is required to ensure precise acquisition [17,18]. On the other hand, AC-EOG is a DC-EOG which has been augmented with a band-pass filtered DC-EOG and is strongly against human movements. The signals are characterized to automatically adjust to zero so that the amount of eye movement is calculated with the signal integral [19].

However, the AC-EOG-based gaze estimation requires calibration to the signal before implementation. The user is required to gaze at a number of target points, and each signal is used for determining the conversion parameters from eye moment and gaze estimation at the calibration phase. The calibration is the most important factor, since it adjusts individual differences caused by electrode attachment, such as plus-shaped five-electrode arrangement and cross-shaped four-electrode arrangement [20,21]. K. Sakurai et al. [16] proposed gaze estimation calibration improvement by combining EOG and Kinect. M. Yan et al. [20] proposed a fuzzy mathematical model to improve the gaze estimation.

In another notable example, Ilhamdi et al. [22] studied the application of AC-EOG gaze estimation for robot control. Four electrodes were attached around the user’s eyes in the cross-shaped arrangement and the calibration was made using affine transformation based on the 24-point-group gaze target. The method was well-converted from two EOG signals to a gaze point on the gaze target. However, the transformation was performed for each quadrant, rather than simultaneously. Then, the quadrant method assumed all the data in one set in space are of the same polarity. However, the significant rotation could disrupt the data polarity. Thus, each quadrant had different characteristics, and data polarity disruption by rotation led to low accuracy.

In this paper, we focused on improving the accuracy of AC-EOG gaze estimation by proposing an alternative 24-gaze-point affine transformation calibration method. A virtual origin was computed which, together with 24 gaze points, formed 25 gaze points which are affine transformed. In this work, we analyzed and compared the accuracy of the calibration technique employed earlier, proposed by Ilhamdi, and the technique proposed in this paper. We applied the proposed method to two traditional AC-EOG signal measurements (i.e., cross-shaped electrode arrangement and plus-shaped electrode arrangement) and demonstrated its superiority in terms of accuracy. The two-electrode arrangements were investigated, as the gaze data were significantly different from each other for the rotation operation.

The accuracy improvement is beneficial for the EOG-based control system. For example, robotic control can pick and place an object using an EOG interfaced robotic arm, while in interface control, EOG can be used in graphical user interface (GUI) navigation.

## 2. Methodology

An experiment to obtain the EOG signal for eye gazing was conducted. The objective was to investigate the accuracy of EOG gazing data with gaze target points. Figure 1 shows the experimental setup for the study. A 24-point target based on a GUI program was built as the target for eye gaze. Target points were represented in pixel value, and reference points served as target points center coordinates (0,0). The relationship between EOG and eye gazing was determined by analyzing the captured EOG data and comparing it with the already-known target pixel value.

The experiments were performed with 10 test subjects (having good visual capabilities and aged between 22 to 40 years old). One after the other were seated with their chin resting on a fixed base, which was set such that the eyes of the subject were horizontal to the origin of the grid on the screen. The distance between the eyes of the test subject and the computer screen was maintained at 35 cm.

### 2.1. Hardware

#### 2.1.1. Electrode Arrangement

The order in which the electrodes are attached to the face of the subject largely affects the shape for the resulting EOG signal. There are two major attachment shapes adopted in research, as shown in Figure 2: One is plus-shaped five-electrode arrangement, and the other is cross-shaped four-electrode arrangement. The plus-shaped type is suitable for recording EOG signals generated by vertical eye movements and horizontal eye movements, since the electrodes are attached in the same direction. Meanwhile, the cross-shaped type has the advantage of a smaller number of electrodes. However, the technique requires appropriate post-processing conversion of EOG signals to gaze position.

#### 2.1.2. EOG Measurement System

We developed a low-cost AC-EOG measurement system as shown in Figure 3. The device consisted of four main components: Disposable electrodes, an EOG measurement circuit (comprising of a differential amplifier, bandpass filter, and an inverting amplifier, all connected in cascade), a data acquisition device (National Instruments (NI) Corporation USB-6008), and a PC running Windows 10 and VC++ 2017. Disposable electrodes were attached around the user’s eye, which captured DC-EOG generated by eye movements. The EOG system is configured as two channels (Ch1 and Ch2). The bandpass filter, having a lower cut-off frequency of 1.06 Hz and upper cut-off frequency of 4.97 Hz, converts the DC-EOG to AC-EOG, which is then amplified before being fed to the data acquisition device. The value of the bandpass filter was set as such to effectively measure the EOG signal. The sampling frequency in the data acquisition device is 1 kHz. Then, the device converts the electrical potential to digital data and transfer to PC program “Microsoft Visual Studio C/C++ 2017”. Finally, the PC program analyzes with gaze estimation.

### 2.2. Software

#### 2.2.1. AC-EOG Discrimination

In this paper, we determined the gaze estimation by analyzing the two AC-EOG channel (Ch1 and Ch2) signals. The gaze estimation refers to the estimation of the point in space where the eyes of the subject are focused. In other words, it is the determination of the x and y coordinates on a two-dimensional plane.

From the signals, we proposed an integral method to analyze the EOG signal. In the integral computation, we included the signal thresholds method for Ch1 and Ch2 signals. The signal threshold served two purposes. The first purpose was to determine the polarity of the integral value. Positive integral was determined if the signal was above the positive threshold (th+) and negative integral was determined if the signal was below the negative threshold (th-). Second, the thresholds were introduced to remove any unwanted residual noises from the integral value, as noises could affect the accuracy of gaze estimation. The equation to obtain the signal integral is derived as

(1)$${{\mathrm{EOG}}_{\mathrm{integral}}}_{ch_{i}}=\left|\overset{}{\underset{th\_p}{\int }}EOG_{Ch_{i}}\left(t\right)dt\right|+\left|\overset{}{\underset{th\_n}{\int }}EOG_{Ch_{i}}\left(t\right)dt\right|$$

(2)$$\mathrm{th}\_p=\left\{t:EOG_{Ch_{i}}\left(t\right)>th+\right\}$$

(3)$$\mathrm{th}\_n=\left\{t:EOG_{Ch_{i}}\left(t\right)>th-\right\}$$

(4)i=1,2

Figure 4 shows an example of AC-EOG analysis for integral value. We also highlighted that eye blinks, either voluntary or involuntary, are counted for integral value.

Then, to determined the gaze estimation for x and y coordinates, the x value is represented by the channel Ch1 integral value and y value from channel Ch2 integral value. The values are used to compare with gaze target coordinates.

#### 2.2.2. EOG-Gaze Target Algorithm

Ideally, the EOG gazing integral value should match the gaze target coordinate, as shown in Figure 5a. Previous studies for the cross-shaped electrode attachment show that there is a significant problem with accuracy between gazing data and targets [21]. Figure 5b shows the estimated gazing data obtained using the cross-shaped electrode arrangement. It can be observed that the estimated coordinates are rotated relative to the axes. Gazing data obtained using the plus-shaped electrodes are more accurate than the cross-shaped electrodes, as seen in Figure 5c.

#### 2.2.3. Coordinate Transformation Method

The difference between gaze data and gaze target coordinates could be corrected using coordinate transformation techniques. A 24-point gazing data calibration method was previously developed by Ilhamdi et al. [22] based on the cross-shaped electrode attachment to improve the gazing data. In this conventional method, the 24-point gazing data are spatially separated into quadrants. The data conversion is then implemented based on the quadrant.

In this paper, we proposed a simpler computation using the affine transformation. Instead of separating the data, we propose a calibration method based on virtual origin coordinates. The conversion based on the proposed technique enables all gazing data to be calibrated simultaneously. Equations (5)–(8) shows the proposed homogeneous matrix for the affine transformation. The gazing data conversion (x’, y’) is determined using Equation (9).

(5)Homogeneous Matrix= [Dilatation][Rotation][Shear][Translation]

(6)$$=\;\left[\begin{array}{ccc} s_{x}\left(\cos\theta -m_{2}\sin\theta \right) & s_{x}\left(m_{1}\cos\theta -\sin\theta \right) & f_{1}\\ s_{y}\left(\sin\theta +m_{2}\cos\theta \right) & s_{y}\left(m_{1}\sin\theta +\cos\theta \right) & f_{2}\\ 0 & 0 & 1 \end{array}\right]$$

(7)$$f_{1}=s_{x}\left(-T_{x}\cos\theta +T_{x}m_{2}\sin\theta -T_{y}m_{1}\cos\theta +T_{y}\sin\theta \right)$$

(8)$$f_{2}=s_{y}\left(-T_{x}\sin\theta -T_{x}m_{2}\cos\theta -T_{y}m_{1}\sin\theta -T_{y}\cos\theta \right)$$

(9)$$\left[\begin{array}{c} x^{\prime }\\ y^{\prime }\\ 1 \end{array}\right]=\mathrm{Homogeneous}\;\mathrm{Matrix}\;\left[\begin{array}{c} x\\ y\\ 1 \end{array}\right]$$

There are four geometrical steps involved in the homogeneous matrix computation. Figure 6 illustrates each geometry processes. The first step is the translation, where the imaginary center coordinate (Tx, Ty) is adjusted to the origin coordinates (0,0). The second step involves shearing of the axes. The shear is to ensure that the axes of the captured gazing data are perpendicular to each other. The third step is to rotate the now perpendicular axes by an angle θ to match normal x–y plane. The final step is dilatation. Dilatation adjusted the gazing data to have similar value with the gaze target pixel value. However, in this part, we separated the dilatation into 4fourvariables based on the line axis. The dilations are determined as:Dilatation s1 is for data located on the positive y-axis line;Dilatation s2 is for data located on the negative y-axis line;Dilatation s3 is for data located on the positive x-axis line;Dilatation s4 is for data located on the negative x-axis line.

Compared to the conventional method, the number and the order of the proposed geometrical steps are significantly different. The proposed method uses one less geometrical step than the conventional method. Besides, the total number of mathematical operation parameters required in the proposed method are nine: Two translation, two shear operations, one rotation, and four dilatation operations. Compared with the conventional method, which required seven operation parameters for each quadrant, giving a total of twenty-eight operation parameters, this shows that the proposed method is simpler than the conventional method in terms of computational resources and time.

## 3. Experiments and Discussion

### 3.1. Experiment 1: Cross-Shaped Electrode Attachment Transformation Method Simulation

A comparative investigation experiment was conducted to compare the accuracy of conventional cross–shaped electrode attachment method conversion method with the proposed method. Four simulation patterns of 24-point gazing data were proposed for the experiment. The patterns were used to test each computation in affine transformation homogeneous matrix: Dilatation, rotation, shear, and translation. To investigate the accuracy, the pixel distance error was used as an indicator, where the lower the distance error value, the higher the accuracy. Figure 7 shows the simulation patterns used in the experiment. The specification for each pattern computation objectives is as such:(1)Pattern (a) is to test dilatation and rotation;.(2)Pattern (b) is to test dilatation, rotation, and translation;(3)Pattern (c) is to test dilatation, rotatin, and shear;(4)Pattern (d) is to test dilatation, rotation, shear and translation.

Figure 8 shows the computation results. Based on the results, the proposed method performed better in calibrating the simulated gazing data relative to the conventional technique. From the four patterns, the average pixel distance error for the x and y pixel was negligibly small for the proposed method. This meant that no errors were observed for the proposed method. In contrast, the average pixel distance error for x pixel was 11±3 pixel and for y pixel , the error was 5±2 pixel for the conventional method. From the results, the proposed virtual origin coordinate conversion method is significantly better than the conventional method.

The errors observed for the conventional method are attributed to quadrant separation-based computation. The quadrant method restricts the view on the calibration into a small set of data. The data near the x and y axis are adversely affecting the computation. The quadrant method assumed all the data in one set in space are of the same polarity. If the rotation is significant, the data polarity is disrupted. The borderline data polarity, when used for computation, became inconsistent and then affected the accuracy of the calculations. In the proposed method, however, the 24-point gazing data are calibrated simultaneously, making the method not only simpler, but also accurate when it comes to computation.

### 3.2. Experiment 2: Accuracy Performance Between Electrode Attachment Methods

In this section, we performed a comparative investigation of the accuracy of the proposed method using EOG gazing data. Ten samples of 24-point gazing data were taken for both electrode attachment methods. We determined the accuracy based on gaze angle error. The lower the angle degree, the more accurate the gazing data. Figure 9 illustrates the original EOG gazing data.

Figure 10 shows the result of cross-shaped electrode attachment. Comparied to the original EOG gazing data, the accuracy is greatly improved. The calibrated 24-point gazing data achieved compelling similarity with gaze targets. Based on the bar chart, most of the x and y error angle for each point is less than 2°. On the other hand, Figure 11 shows the results of plus-shaped electrode attachment. Likewise, compared to the original EOG gazing data, the accuracy is also greatly improved. The gaze error angle observed from bar chart shows that the x and y angle error is less than 2°.

The next step in the experiment is to obtain EOG gazing data from 10 test subjects. The test subjects are to strengthen the investigation on the accuracy of the proposed method. However, instead of observing the gaze angle error on a point-to-point basis, the average for 24-point is calculated. The results for gazing data are shown in Figure 12.

The test subjects’ average gaze angle error results are shown in Figure 13, Table 1 and Table 2. The ten test subjects’ average for the cross-shaped electrode attachment is x=2.27°±0.46° and y=1.83°±0.34°. In comparison, the average for the plus-shaped electrode attachment is x=0.94°±0.19° and y=1.48°±0.27°. From the gaze angle error result, both electrode methods show almost identical gaze angle errors. However, the plus-shaped electrode attachment method shows a relatively small gaze angle error compared to the cross-shaped electrode attachment.

From the experiment, the proposed method proved to increase the accuracy of EOG gazing data for two different electrode attachment methods. However, in terms of the simplicity of the conversion method, the plus-shaped electrode attachment offered simpler computation than the cross-shaped electrode attachment method. The rotation for plus-shaped dive-electrode attachment is negligible. Thus, the cross-shaped electrode attachment has nine operation parameters for affine transformation, but we had eight operation parameters for the plus-shaped electrode attachment.

## 4. Conclusions

The objective of the research was to enhance the accuracy of the EOG gaze estimation method. The AC-EOG signal method was used to estimate eyeball movement and the gaze position was determined by converting the captured signals. In conventional research, the affine transformation was introduced as a calibration method for 24-point gazing data from the two AC-EOG signals, corresponding to vertical and horizontal eyeball movements using the cross-shaped electrode attachment method. However, the transformation was not applied for all the 24-point gazing data, but was instead applied for four spatially separated data (quadrant method). The quadrant method was also easily influenced by data rotation. The quadrant method assumed all the data in one set in space are of the same polarity, however, the significant rotation could disrupt the data polarity. The influence of data rotation was not fully investigated, as one electrode attachment method (cross-shaped four-electrode attachment) was used. Then, in term of the computational complexity of the quadrant method, seven variables were computed in each quadrant, where each quadrant produced a different value. In this paper, we proposed the conversion method for 24-point gazing data simultaneously and assumed a virtual origin (i.e., 25th point) on gaze coordinates with 24-point gazing data and applied an affine transformation to 24-point gazing data. Two experiments were conducted as comparative investigation for the conventional and proposed methods. The first experiment was an accuracy investigation between the proposed method and conventional computation. Four simulation patterns, based on 24-point gazing data, were used, and the accuracy was determined by the pixel distance error. The result shows that the proposed method achieved negligible error in gazing data conversion. The second experiment was to determine the accuracy of the proposed method using EOG gazing data. Ten test subjects were used to performed 24-point gaze targets with two different electrode attachment methods. The average angle error for the cross-shaped electrode attachment was x=2.27°±0.46° and y=1.83°±0.34°. On the other hand, the plus-shaped electrode attachment had an error of x=0.94°±0.19° and y=1.48°±0.27°. The results show that there was minimal error using the proposed method, and the two electrode attachment methods resulted in almost identical performances. From the experiments, we conclude that the proposed method was simpler and more accurate in EOG gaze estimation than the conventional method.

## Figures and Tables

**Figure 1 sensors-19-03650-f001:**
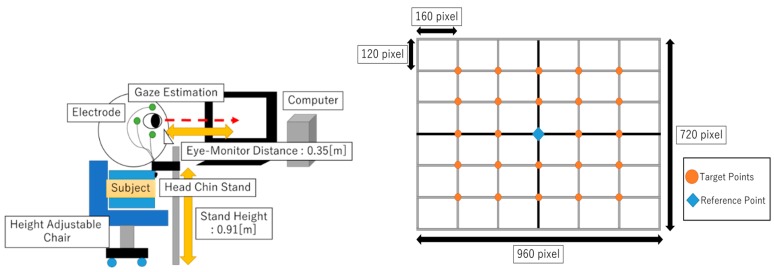
Experiment setup (**left**) and 24-point gaze targets (**right**).

**Figure 2 sensors-19-03650-f002:**
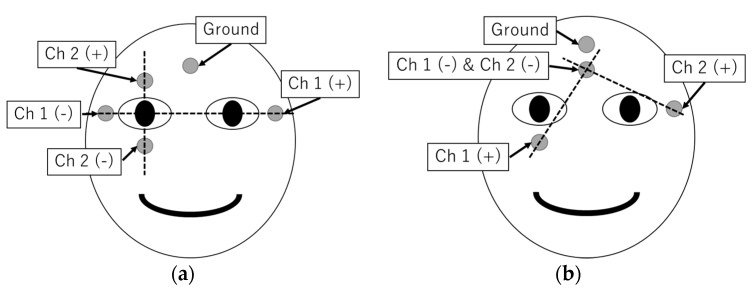
(**a**) Plus-shaped electrode attachment and (**b**) cross-shaped electrode attachment.

**Figure 3 sensors-19-03650-f003:**
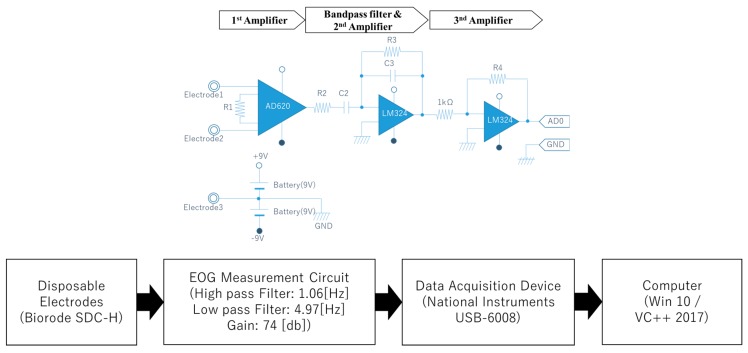
Electro-occulo-graphic (EOG) measurement device and the processes.

**Figure 4 sensors-19-03650-f004:**
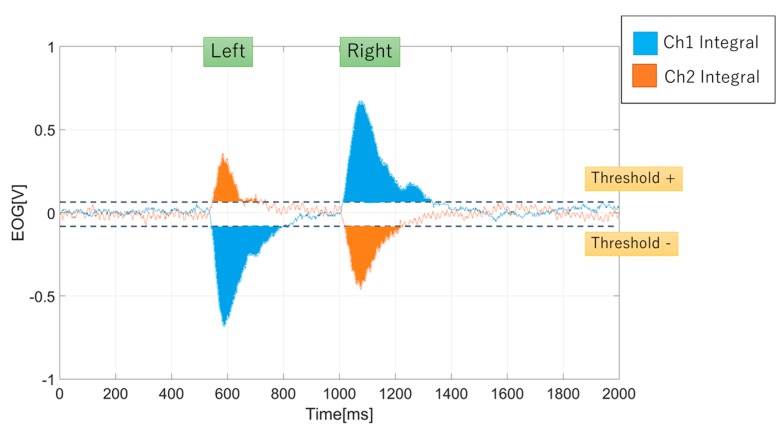
Signal integral method for left and right eye movement.

**Figure 5 sensors-19-03650-f005:**
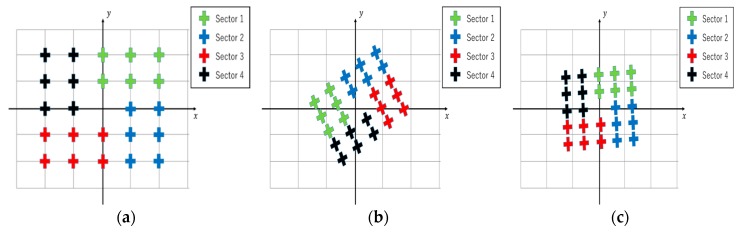
Gaze estimation: (**a**) Ideal; (**b**) Cross-shaped-electrode attachment; (**c**) Plus-shaped-electrode attachment.

**Figure 6 sensors-19-03650-f006:**
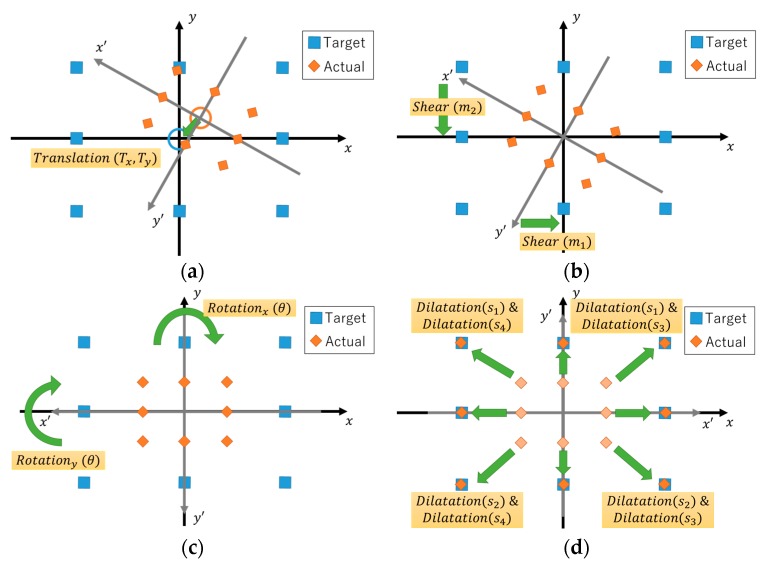
Proposed affine transformation geometry process. (**a**) Translation; (**b**) Shear; (**c**) Rotation; and (**d**) Dilatation.

**Figure 7 sensors-19-03650-f007:**
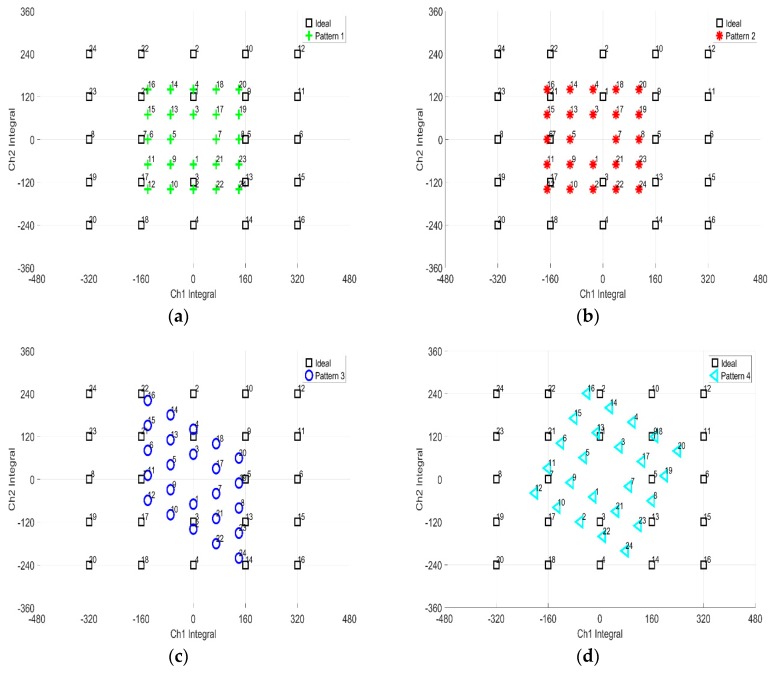
Simulation patterns for 24-point computation: (**a**) Dilatation and rotation; (**b**) Dilatation, rotation and translation; (**c**) Dilatation, rotation and shear; (**d**) Dilatation, rotation, shear and translation.

**Figure 8 sensors-19-03650-f008:**
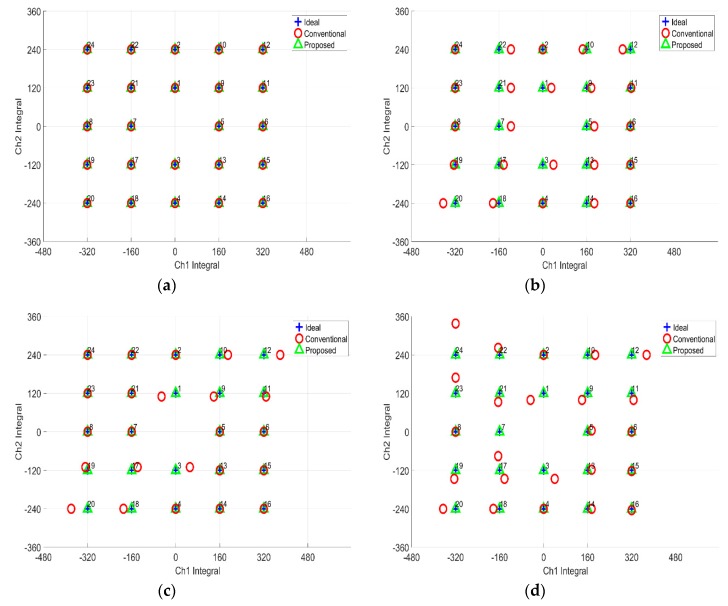
Simulation results for 24-point computation: (**a**) Dilatation and rotation; (**b**) Dilatation, rotation and translation; (**c**) Dilatation, rotation and shear; (**d**) Dilatation, rotation, shear and translation.

**Figure 9 sensors-19-03650-f009:**
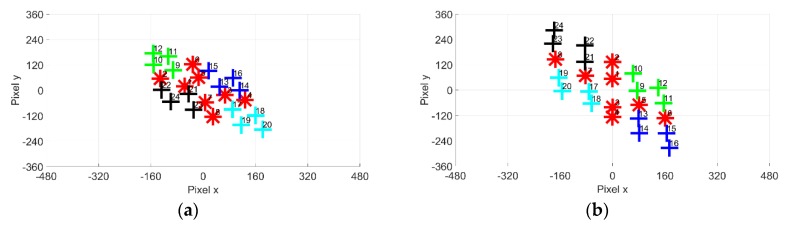
Original EOG gazing data: (**a**) Cross-shaped electrode attachment; (**b**) Plus-shaped electrode attachment.

**Figure 10 sensors-19-03650-f010:**
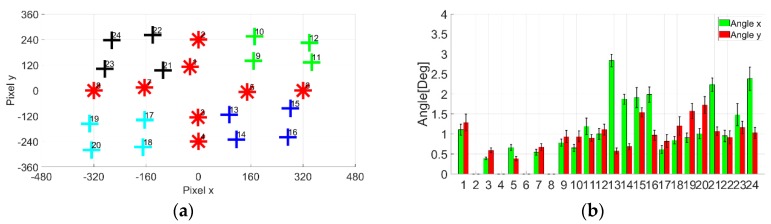
Cross-shaped electrode attachment conversion results: (**a**) Calibrated 24-point gazing data; (**b**) Gaze angle error for each point.

**Figure 11 sensors-19-03650-f011:**
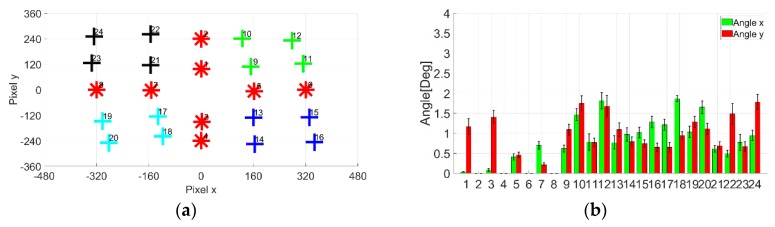
Plus-shaped electrode attachment conversion results: (**a**) Calibrated 24-point gazing data; (**b**) Gaze angle error for each point.

**Figure 12 sensors-19-03650-f012:**
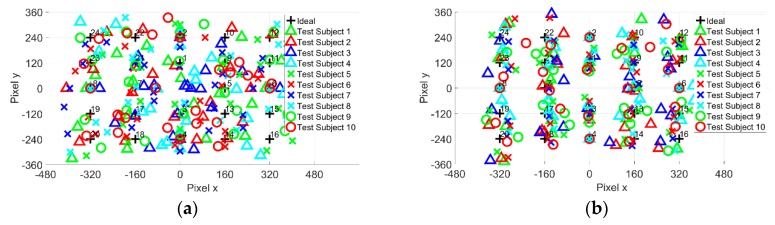
Twenty-four-point gazing data for 10 test subjects (**a**) Cross-shaped electrode attachment; (**b**) Plus-shaped electrode attachment.

**Figure 13 sensors-19-03650-f013:**
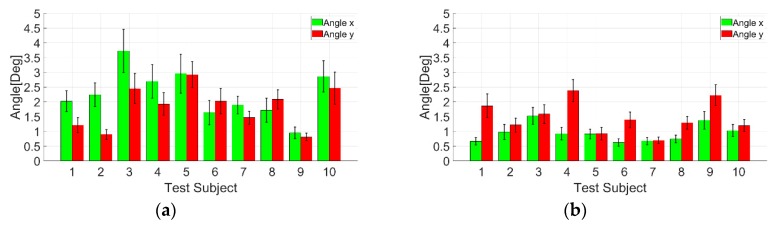
Average 24-point gaze angle error for 10 test subjects: (**a**) Cross-shaped electrode attachment; (**b**) Plus-shaped electrode attachment.

**Table 1 sensors-19-03650-t001:** Cross-shaped electrode attachment average 24-point gaze angle error for 10 test subjects.

Subject	Ave. Angle x (θ°)	Ave. Angle y (θ°)	$x\;\mathrm{SD}\;\left(\sigma_{x}{}^{\circ}\right)$	$y\;\mathrm{SD}\;\left(\sigma_{y}{}^{\circ}\right)$
**1**	2.0205	1.2098	0.3555	0.2517
**2**	2.2390	0.8978	0.4006	0.1682
**3**	3.7267	2.4537	0.7384	0.5082
**4**	2.6903	1.9262	0.5695	0.3820
**5**	2.9605	2.9202	0.6591	0.4417
**6**	1.6390	2.0228	0.4146	0.4356
**7**	1.8961	1.4659	0.2978	0.2190
**8**	1.7187	2.0854	0.4073	0.3257
**9**	0.9506	0.8094	0.1957	0.1324
**10**	2.8613	2.4653	0.5350	0.5427
**Average**	2.2703	1.8256	0.4574	0.3407

**Table 2 sensors-19-03650-t002:** Plus-shaped electrode attachment average 24-point gaze angle error for 10 test subjects.

Subject	Ave. Angle x (θ°)	Ave. Angle y (θ°)	$x\;\mathrm{SD}\;\left(\sigma_{x}{}^{\circ}\right)$	$y\;\mathrm{SD}\;\left(\sigma_{y}{}^{\circ}\right)$
**1**	0.6627	1.8633	0.1280	0.4032
**2**	0.9768	1.2166	0.2554	0.2388
**3**	1.5283	1.5873	0.2783	0.3169
**4**	0.9221	2.3825	0.2106	0.3763
**5**	0.9128	0.9252	0.1598	0.2105
**6**	0.6242	1.3943	0.1284	0.2691
**7**	0.6691	0.6944	0.1211	0.1133
**8**	0.7453	1.2921	0.1303	0.2095
**9**	1.3774	2.2247	0.2976	0.3514
**10**	1.0307	1.1971	0.2007	0.2041
**Average**	0.9449	1.4777	0.1910	0.2693

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
