# Peer review of "24-Gaze-Point Calibration Method for Improving the Precision of AC-EOG Gaze Estimation"

_sensors, 2019, doi:10.3390/s19173650_

Round 1
Reviewer 1 Report
The paper from dr. Suhaimi (24-Gaze-Point Calibration Method for Improving the Precision of AC-EOG Gaze Estimation) is presenting an improved calibration method, based on a 24-gaze-point procedure, for gaze estimation. The paper is correctly structured and appropriately written in English (minor English language issues are present).
Some important aspects have been individuated and should be corrected before publication:
1. The novelty aspects of the proposed method are not clearly individuated and put in relation to the present state-of-art in the EOG field. In particular the accuracy of the available/published methods are not reported, and the reader can not compare these available methods with the one proposed by the authors. Such aspects should be reported in the Introduction paragraph and discussed in depth in the Conclusion.
2. The number of experimental tests is relatively low (n=4). I recommend reaching a number of subjects to be tested of 10– at least. This, even if not significant from the statistical point of view would allow to have a mean and SD value to discuss.
3. The Introduction paragraph is missing the application field of the proposed method. Where the proposed method is applied (ophthalmology, industry, robotics, assistive technologies?) and what are the requirements (i.e. accuracy) provided by the different operative contexts?
4. (line 39) – Report what is the ‘preciseness’ (please change the term in ‘accuracy’) of the EOG based eye tracker system.
5. The plus-shaped 5 electrodes, the cross-shaped 4 electrode arrangement should be proposed and put in relation with the electrodes set-up used by the authors. What are the differences?
6. (line 60) – It is not clear why the Ilhamdi method – not presented and discussed with sufficient detail – is ‘…characterized less preciseness’. Please provide more detailed information of this method, being used as benchmark respect to the author’s proposed approach.
7. In figure 1 and in the text (line 77) it is reported that the subject head to the PC screen is 35 cm. What part of the head (nose, eye, electrodes?). Are the different distances of the 24-point taken in consideration in the calibration method?
8. It is not clear why two electrodes configurations are studied. What are the differences between them in terms of pros/cons?
9. (line 97) – why the filter bandwidth has been chosen as indicated (1.06 – 4.97 Hz)?
10. In paragraph 2.1.1 some relevant information are missing: sampling frequency? Number of acquired channels? Trigger?
11. In figure 3, please remove the photo of the system (it is not clear and do not provide useful information).
12. (line 108) – The title of the paragraph is not clear. What is the significance of the AC-EOG Integral? Please change it or provide more information to the reader to understand what its meaning is. Paragraph 2.2.1 is not sufficiently clear, please re-write it.
13. (line 163/4) – why ‘…the order of the proposed geometry processes are significantly different.’? In what are different: less, more, simpler?
14. (line 167) – Please justify the sentence: ‘This shows that the proposed method is simpler than the conventional method.’.
15. (line 199/200) – Please justify the sentence: ‘From the experiment, the proposed imaginary center coordinates conversion method is significantly better than the conventional method.’.
16. (line 276) – The acknowledgment should be addressed to the Malaysia Scholarship Organization, Yayasan Pelajaran MARA if this organization was involved in the research.
Minor issues:
a. Please change the term ‘preciseness’, please change it with the more appropriate: ‘accuracy’. Check along the paper.
b. Please change ‘ more simple’ with ‘simpler’. Check along the paper.
c. (line 164) – Please chance ‘…the order of the proposed geometry processes are significantly different.’ In: ‘…the order of the proposed geometry processes is significantly different.’.
Author Response
Thank you very much for taking the time to review our paper and also for giving us a chance to resubmit the revised paper. We believe that all the suggestions and comments raised by you and the reviewers have been carefully considered and fully answered. We are glad to resubmit this revised paper to you. We are pleased to respond to any further comments and questions on the revision. The detailed responses to the Technical Editor and Reviewers suggestions and comments are in the attachment file. Please kindly refer to the attachment.

Reviewer 2 Report
Dear Authors,
Authors propose a method to improve precision of gaze estimation. It sound interesting!!
The manuscript was written easily and logically well. But I think it would be a better paper if you had objective performance evaluations compared to the more recent studies. We've also used all of the resulting comparisons as figures, but it's a good idea to compare the important information in tables, and to do a statistical experiment together. Also, I hope you to improve your English expressions.
Author Response

(The authors gave the same response as above.)

Reviewer 3 Report
The innovation of this study is to investigate on improving the precision of AC-EOG gaze estimation. After careful reading of this manuscript, I have the following comments:
1. This study has its novelty.
2. The manuscript is well written.
3. Authors only fixed the distance between the subject’s face and the computer (0.35m) but they didn’t fix the level (or the height position) of the eye to the computer. This will introduce problem on the data collection.
4. What is the body height of the subjects? Please provide average height with standard deviation.
5. Repeatability test and test-retest should be conducted.
6. Subject inclusion and exclusion criteria should be mentioned.
Author Response

(The authors gave the same response as above.)
